# Evaluation of Changes in the Permeability Characteristics of a Geotextile–Polynorbornene Liner for the Prevention of Pollutant Diffusion in Oil-Contaminated Soils

Jeongjun Park 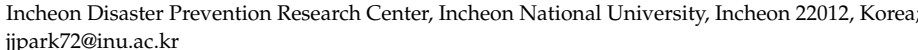

Incheon Disaster Prevention Research Center, Incheon National University, Incheon 22012, Korea; jjpark72@inu.ac.kr

**Abstract:** In this study, changes in the permeability characteristics of a geotextile–polynorbornene liner at different oil pollutant contact times were evaluated. Experiments and numerical analyses were performed, and ASTM D5887 and ASTM D6766 were applied as test methods. The test results show that, when the pollutant contact time and pressure head were 4 h and 75 kPa, the reaction between the geotextile–polynorbornene liner and the pollutant was almost complete. Moreover, a numerical analysis was used to measure the ratio of the concentration of the pollutant that permeated through the geotextile–polynorbornene liner to the initial pollutant concentration at different pollutant contact times. The ratio was between 70 and 83% after a pollutant contact time of 0.5 h and between 0.1 and 1.0% after 4 h. The test and numerical analysis results confirm that, as a reactive medium, the geotextile–polynorbornene liner can effectively prevent the diffusion of oil pollutants by changing its permeability characteristics.

**Keywords:** oil-contaminated soils; geotextile–polynorbornene liner; pollutant adsorption; diffusion; permeability alteration





## 1. Introduction

The rapid growth of the urban population has resulted in higher population densities, enhanced urbanization and industrialization, and increased anthropogenic inputs in the environment, which may threaten sustainable development and worsen several environmental problems, including groundwater and soil pollution [1–3]. Specifically, as a result of increased industrialization in South Korea, higher concentrations of chemicals and increased waste generation from industries have become national concerns because of their negative effects on the soil matrix. Thus, to protect the population from exposure to soil contaminants, several countries have established soil quality standards (SQS) and environmental impact assessments (EIA) for evaluating and monitoring SOC development projects. In particular, some of the most disastrous effects on the soil matrix are the result of oil and chemical pollution, as these pollutants have short- and long-term consequences on the ecosystem and soil makeup. The significant growth in oil and chemical consumption has resulted in increased concentrations of these pollutants in the soil due to leaks and spills from oil storage tanks in gas stations and chemical storage facilities, as well as pipeline ruptures, well blowouts, anthropogenic inputs, and transport accidents. Specifically, total petroleum hydrocarbon (TPH) is frequently released to the environment through accidents in commercial and private facilities and from storage facilities in military bases and industrial complexes. Ławniczak et al. [4] reported that crude oil-based hydrocarbons constitute the largest class of environmental pollutants in the world. With damage at such large scales, many remediation methods, treatment plans, and control strategies are costly and difficult to implement.

In the past years, various remediation techniques have been used to restore contaminated soils using eco-friendly approaches at a relatively low cost. These methods are

divided into ex situ (presence of excavation) and in situ (absence of excavation) treatments, depending on the characteristics of the location, the nature of the pollutants, the degree of pollution, and the types and characteristics of the pollutants in contaminated soils. Ex situ treatment is a remediation strategy that involves the physical removal of certain sites of contamination to another area, preferably within the same location. On the other hand, in situ treatment methods remediate contaminated soils at the original location without excavation [5,6]. These remediation technologies restore contaminated soils. However, it takes considerable time to identify contaminated soil after a polluting event. By the time it is identified, extensive damage has already occurred due to the diffusion of pollutants. Therefore, rather than applying remediation techniques after contamination, the application of proactive treatments at sites where the leakage of pollutants can be reasonably anticipated (gas stations, oil storage facilities, and industrial complexes) can significantly prevent the diffusion of pollutants and reduce the scale of damage. Therefore, researching technology that can prevent the diffusion of pollutants and restore contaminated areas is essential.

Many studies have been conducted on the remediation of oil-contaminated soils. Jeong et al. [7] artificially contaminated soils with different amounts of oil and used TPH analysis to evaluate the effects on the soil composition. On the other hand, Lee et al. [8] applied land farming and high-temperature thermal desorption as a remediation method for petroleum hydrocarbon-contaminated soils and evaluated the pollutant removal efficiency. In addition, Cho et al. [9] researched a mechanism of pollutant removal from TPH-contaminated soils using microwave heating. Sayed et al. [10] reviewed several previous studies on the application of bioremediation to environments contaminated with crude oil, TPH, and related petroleum products. Han [11] evaluated the removal efficiency of a biopile when it was used to restore soils that had been contaminated with low-concentration TPH for 100 days.

The main methods used to remediate oil-contaminated soils are chemical oxidation, which oxidizes pollutants into water and carbon dioxide using an oxidizing agent, and soil washing, which removes pollutants through contact between an aqueous solution containing a cleaning agent and contaminated soils (Feng et al. [12]). Lee et al. [13] used soil washing as a method to reduce the pollutant concentration in oil-contaminated soils and evaluated its efficiency in removing TPH from diesel-contaminated sand. Previous studies related to the remediation of oil-contaminated soils using soil washing have mainly used nonionic and anionic surfactants as cleaning agents. Khalladi et al. [14] reported that surfactants were effective in removing TPH adsorbed on the surface of soil particles by reducing the interfacial tension between the soil particles and oil. Vreysen and Maes [15] used sandy loam that was artificially contaminated with diesel and reported that nonionic surfactants had a removal efficiency of 50%. In addition, Hernández-Espriú et al. [16] reported that soil washing could achieve a TPH removal rate of 60%. Jang et al. [17] used plasma blasting for the remediation of contaminated soils and demonstrated the applicability of the technique by evaluating the fluid diffusion effect, the improved permeability of the contaminated soils, and the purification efficiency.

The diffusion of pollutants is caused by concentration changes that occur in the liquid state. Previous studies have suggested that this phenomenon can be prevented by applying a reactive medium that reduces the concentrations of solid (e.g., heavy metals) and liquid pollutants (e.g., oil) in groundwater. In particular, contaminated groundwater can be remediated using permeable reactive barriers (PRBs), which utilize effective, eco-friendly, and cost-efficient reactive media, as well as appropriate construction methods for the site [18–21]. PRBs are generally installed in the ground where contaminant plumes exist and then use their hydraulic flow. They remove pollutants by inducing physicochemical and biological reactions between reactive media and pollutants [18,22].

Moreover, many studies have been conducted on liner systems, PRBs, reactive barriers, and reactive media to prevent the diffusion of pollutants. In particular, geosynthetic clay liners (GCLs), in which the bentonite layer is surrounded by geotextiles or geomem-

branes, have been widely distributed to prevent the diffusion of pollutants in fluids. GCLs have been applied to many geotechnical fields, including landfills, dams, artificial lakes, sewage treatment ponds, storage tanks, and contaminated soils. The popularity of PRBs is due to the variety of advantages that they offer, such as low permeability coefficients, low hydraulic conductivity, high mechanical stability, and simple and rapid on-site installation [23–27]. Xue et al. [28] conducted permeability tests on GCLs soaked in various types of solutions with different concentrations and analyzed the relationship between the expansion and permeability coefficient of GCLs.

Kim and Lee [29] evaluated the treatment efficiency of groundwater contaminated with heavy metals by applying zero-valent iron, steel slag, activated carbon, and tree bark to PRBs as reactive media. Ji and Cheong [30] applied PRBs as a method for remediating contaminated leachate from mines on-site and recommended the application of organic carbon mixtures as reactive media to remove high concentrations of aluminum. Furthermore, Chung and Lee [31] evaluated the suitability and limitations of Moringa oleifera mass bentonite (MOM-bentonite) as the reactive media of reactive barriers for treating aquifers contaminated by the movement of PCE-contaminated groundwater on site. Guerin et al. [32] evaluated the applicability of PRBs for the remediation of groundwater contaminated with petroleum hydrocarbons. Moreover, Cho et al. [33] evaluated the applicability of pyrophyllite as a reactive medium for PRBs to prevent the diffusion of pollutants in contaminated groundwater and reduce the environmental pollution of soils. Kim et al. [34] evaluated the concentration of solidifying agents containing fly ash and lime as well as the optimum water content for the formation of mixed barriers to purify contaminated groundwater in soils classified as SW-SC. They also evaluated the performance of a mixed liner and cover materials containing solidifying agents. In addition, Yun et al. [35] conducted a compaction test on calcium bentonite–sand mixtures to determine the optimum water content. They also evaluated the permeability characteristics of the mixed liner and cover materials by conducting variable-head permeability tests according to the mixing ratio of calcium bentonite.

Incineration or recovery methods applied after using various oil-absorbing materials are widely known treatments for oil-contaminated soils [36–38]. To prevent soil contamination from oil spills, fabric-based oil absorbents have been primarily used for ground surfaces. Non-woven fabrics that use hydrophobic hydrocarbon-based fibers have been most frequently utilized [39,40]. In recent years, oil-absorbing resins that use various adsorption or gelation-type polymers have been increasingly studied [41–44]. Jeong [45] evaluated the oil adsorption characteristics of polypropylene (PP) materials treated with a lipophilic acrylic resin. Gelling agents have a high rate of reaction with oil, and the substances generated in the reaction can be easily recovered. Therefore, Yun et al. [46] applied a mixture of calcium bentonite and a gelling agent as a liner and cover material and evaluated its permeability characteristics after it reacted with trichloroethylene (TCE)—a dense non-aqueous phase liquid (DNAPL) substance—in contaminated soil. Nguyen et al. [47] (2021) evaluated adsorption materials containing polydimethylsiloxane (PDMS) for the removal of oil, and Taylor et al. [48] evaluated the performance of materials that can adsorb pollutants from hydrocarbon-contaminated groundwater.

As mentioned above, many studies have been conducted to prevent the diffusion of pollutants in oil-contaminated soils or to remove such pollutants. However, most of the studied technologies are applied after the occurrence of pollution accidents. Therefore, in this study, a geotextile–polynorbornene liner was used as the oil-absorbing material in reactive barriers to instantly prevent the diffusion of oil pollutants in soils in the event of an oil spill in facilities where such accidents may occur. As the permeability performance is the most important factor in preventing the diffusion of pollutants, the applicability of the geotextile–polynorbornene liner and cover material was evaluated based on changes in its permeability characteristics when it contacts the oil pollutant. The applicability was evaluated using experimental and numerical analyses.

## 2. Overview of the Geotextile–Polynorbornene Liner

In general, bentonite minerals that constitute GCLs selectively adsorb moisture and swell. When bentonite particles that exhibit swelling behavior above a certain level are constrained using upper and lower fabric layers, the GCLs become impermeable. In other words, if a synthetic resin that adsorbs oil replaces bentonite to prevent the diffusion of oil pollutants, it exhibits the same behavior as that of GCLs. This concept is illustrated in Figure 1. The barrier is formed by using upper and lower geosynthetic layers to constraining an oil-absorbing synthetic resin that reacts only with oil; therefore, before an oil spill, the groundwater flows normally because water does not react with the synthetic resin. However, in the event of an oil spill on the ground, the absorption, swelling, and gelation of the oil-absorbing synthetic resin occur when it contacts the oil. In addition, because an impermeable layer is formed by the chemical reaction of the synthetic resin, it is possible to prevent the diffusion of oil. In this study, polynorbornene, which has excellent gelation properties, was applied as a reactive material to oil, and a geotextile was applied as a geosynthetic liner that constrains polynorbornene. Therefore, the oil-absorbing material was named the geotextile–polynorbornene liner.

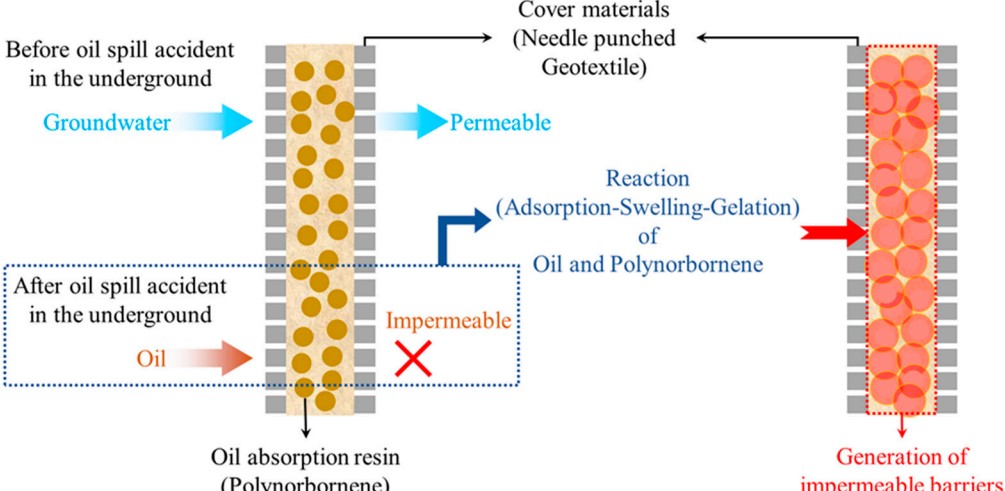

**Figure 1.** The generation of impermeable barriers of the geotextile–polynorbornene liner.

If the geotextile–polynorbornene liner is applied as a reactive medium to form reactive barriers such as PRBs. Then, in the event of an oil spill, the reaction of the material can prevent the diffusion of oil pollutants. Figure 2 shows a conceptual diagram of the prevention of oil pollutant diffusion.

Figure 3 shows the morphology of polynorbornene powder at $100\times$ magnification. Polynorbornene powder particles have a very irregular geometry. To examine the degree of adsorption and swelling of the polynorbornene powder, a simple test on the change in state was conducted, as shown in Figure 4. It can be inferred from the figure that 24 h after mixing polynorbornene powder with diesel, the weight and volume increased as the powder reacted with and absorbed the oil. This indicates that polynorbornene powder can be utilized as an impermeable material through its gelation.

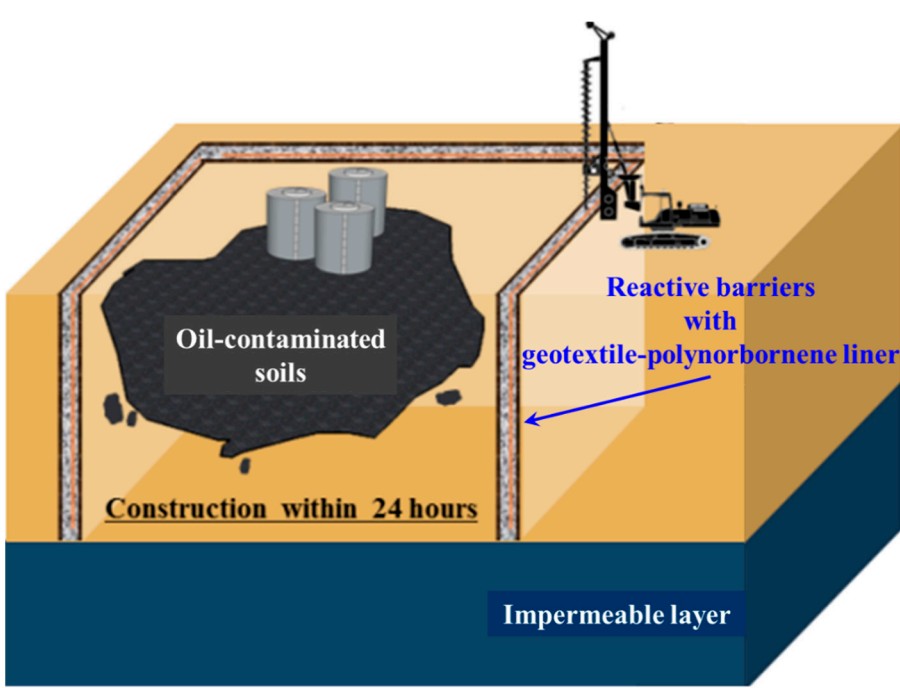

**Figure 2.** Conceptual diagram of reactive barriers with the geotextile–polynorbornene liner.

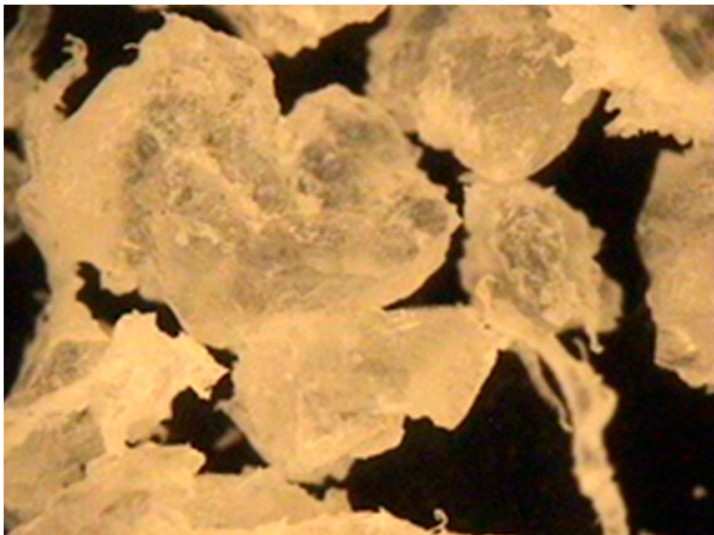

**Figure 3.** Morphology of polynorbornene powder at $100\times$ magnification.

The impermeability performance of polynorbornene after completing gelation must be confirmed before applying it as a liner. From the perspective of geoenvironmental engineering, the impermeability performance of a material is evaluated using its permeability coefficient, which is defined as $10^{-7}$ cm/s or less for a typical impermeable layer. As it is necessary to evaluate the permeability coefficient of the geotextile–polynorbornene liner over time, experiments were conducted in this study to measure changes in its permeability characteristics over time.

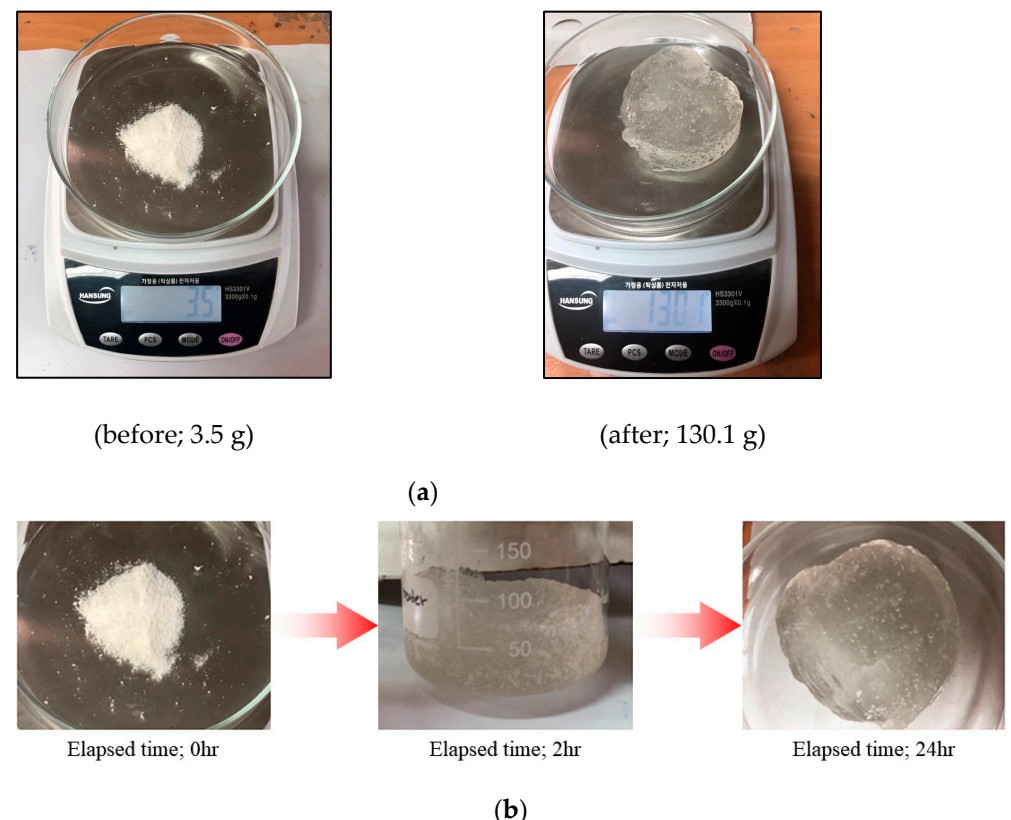

**Figure 4.** Adsorption and expansion of polynorbornene powder: (**a**) weight change after reaction with diesel; (**b**) gelation over time.

## 3. Materials and Methods

### 3.1. Test Apparatus and Materials

To evaluate the permeability of the geotextile–polynorbornene liner, a permeability coefficient similar to that of typical soils in the absence of pollutants was employed with a range of $10^{-2}$ to $10^{-4}$ cm/s. Soil that comes into contact with pollutants has a typical impermeability of $10^{-7}$ cm/s or less.

There are several test methods used to evaluate the permeability performance of materials, but methods based on reactions between pollutants and the liner and cover materials are limited. Conventionally, permeability tests for typical liners and cover materials such as GCLs have been conducted using water after swelling for ≥48 h. Furthermore, the impermeability performance of the materials against oil pollutants with varying concentrations must also be assessed. Therefore, test methods that consider these conditions were used in this study. Two American Society for Testing and Materials (ASTM) International methods for evaluating the permeability and impermeability performances of liners and cover materials were adopted: ASTM D5887 is the standard test method for measuring the index flux through saturated geosynthetic clay liner specimens using a flexible wall permeameter, and ASTM D6766 is the standard test method for evaluating the hydraulic properties of geosynthetic clay liners permeated with potentially incompatible aqueous solutions. In addition, a test apparatus that can evaluate changes in the permeability characteristics of the geotextile–polynorbornene liner according to its contact time with the oil pollutant was also adopted. As shown in Figure 5, the test apparatus consists of a water and air controller, a pressure controller, and an upper/lower pressure cell controller.

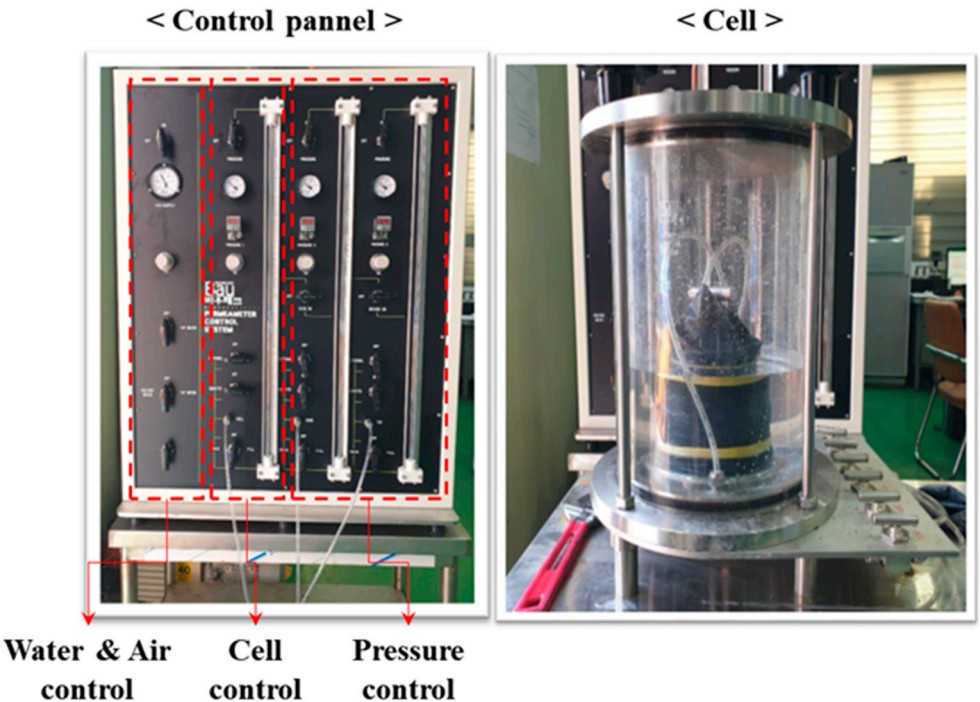

**Figure 5.** Test apparatus used for permeability performance evaluation.

Figure 6 shows the cross-section of the geotextile–polynorbornene liner used in the test. This material has a non-woven fabric made of PP and polyethylene (PET) at the top and a woven fabric made of PP at the bottom. In addition, polynorbornene powder was located between the non-woven fabric (top) and woven fabric (bottom). Lastly, diesel was used as an oil pollutant to induce a reaction with the geotextile–polynorbornene liner.

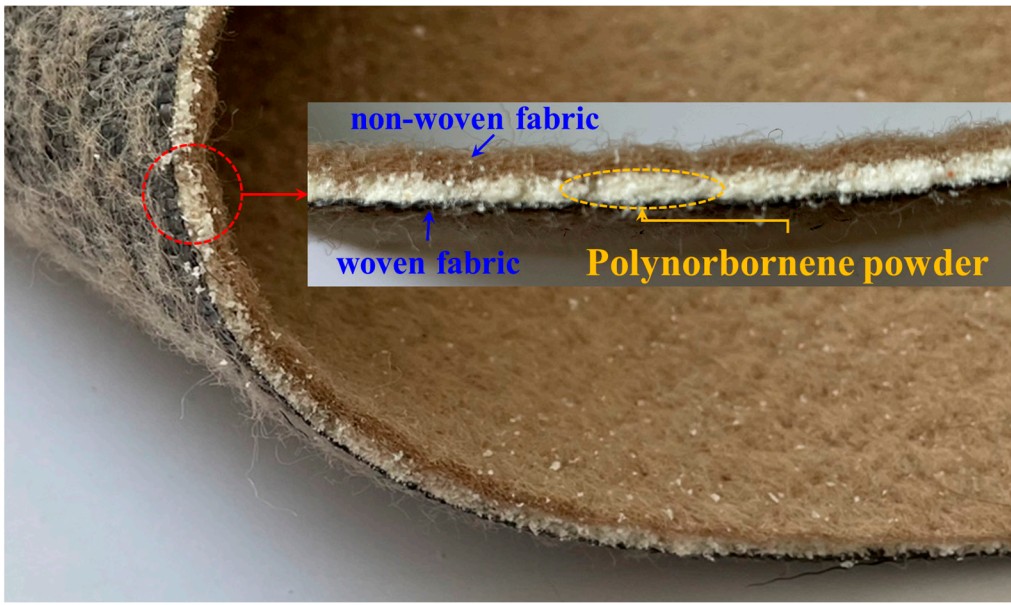

**Figure 6.** Cross-section of the geotextile–polynorbornene liner.

### 3.2. Test Procedure

Changes in the permeability characteristics of the geotextile–polynorbornene liner were tested using the following procedure, in accordance with the methods of ASTM D5887 and ASTM D6766: (1) 100 mm diameter circular samples (geotextile–polynorbornene liner) were prepared; (2) the sample holder was installed at the bottom inside the cell, the lower

porous plate was installed, the sample was placed in the holder, and the upper porous plate was installed; (3) the membrane and O-ring were installed to prevent the leakage of the oil pollutant (diesel) during the reaction; (4) the cell pressure (35 kPa) and upper/lower back pressure (7–14 kPa) were established; (5) the cell and back pressures were increased every ten minutes; (6) the final pressures (cell pressure = 550 kPa, back pressure = 515 kPa) were established; (7) the sample was stabilized under pressure for 40 h; (8) a pressure head of 15 kPa was generated after setting the lower back pressure to 530 kPa to cause the upward penetration of the pollutant; (9) the burette reading was recorded over time after inducing the permeation of the oil through the sample; and (10) the flux and permeability coefficient were calculated. The procedure is summarized in Figure 7.

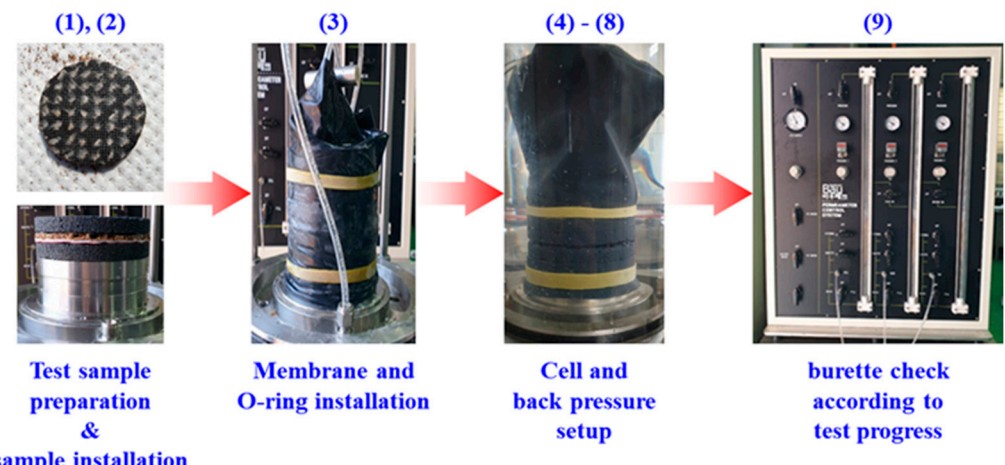

**Figure 7.** Test procedure.

The flux was obtained to calculate the permeability coefficient. The flux is the laminar flow per unit volume, that is, the flow of water that passes through the cross-section of the sample per unit time, and it is expressed in units of velocity. The flux calculation results were used to determine the permeability coefficient, which is defined as the laminar flow that passes through the cross-section per unit time under the influence of a hydraulic gradient and is expressed in units of velocity. Therefore, the permeability coefficient can be calculated based on the flux using Equations (1)–(4).

$$F = \frac{Q}{A} \tag{1}$$

$$V = ki \tag{2}$$

$$i = \frac{\Delta h}{L} \tag{3}$$

$$\Delta h = \frac{\Delta P}{\rho g} \tag{4}$$

where $F$ is the flux (m$^3$/m$^2$·s), $Q$ is the flow rate (m$^3$/s), $A$ is the cross-sectional area (m$^2$), $V$ is the discharge velocity (cm/s), $k$ is the permeability coefficient (cm/s), $i$ is the hydraulic gradient, $L$ is the specimen length (m), $\Delta P$ is the pressure head (kPa), $\Delta h$ is the total head (m), $\rho$ is the density of water (ton/m$^3$), and $g$ is the gravitational acceleration.

Table 1 lists the test conditions used in the study. The pollutant contact time started at 0 h (before contact with the pollutant) and was monitored 0.5, 1, 2, 4, 16, and 24 h after initial contact with the pollutant. The pressure head ranged between 15 and 45 kPa (three times), 75 kPa (five times), and 105 kPa (seven times) to analyze the discharge time. In addition, the discharge time was set according to the pressure head to ensure a constant flow for each pollutant contact time.

**Table 1.** Test conditions.

| Contact Time of Oil Pollutant | Pressure Head ($\Delta P$, kPa) | Total Head ($\Delta h$, m) | Discharge Time ($t$, s) | Flow Rate ($Q$, cm$^3$/s) | Specimen Area ($A$, cm$^2$) | Specimen Length ($L$, cm) |
|---|---|---|---|---|---|---|
| 0 h | 15 | 1.53 | 8.2 | 40 | | |
| | 45 | 4.59 | 3.8 | | | |
| | 75 | 7.65 | 2.8 | | | |
| | 105 | 10.71 | 2.3 | | | |
| 0.5 h | 15 | 1.53 | 16 | | | |
| | 45 | 4.59 | 13 | | | |
| | 75 | 7.65 | 12 | | | |
| | 105 | 10.71 | 9 | | | |
| 1 h | 15 | 1.53 | 540 | | | |
| | 45 | 4.59 | 94 | | | |
| | 75 | 7.65 | 34 | | | |
| | 105 | 10.71 | 15 | | | |
| 2 h | 15 | 1.53 | 780 | | 50.27 | 2 |
| | 45 | 4.59 | 110 | | | |
| | 75 | 7.65 | 57 | | | |
| | 105 | 10.71 | 29 | | | |
| 4 h | 15 | 1.53 | 8242 | 0.5 | | |
| | 45 | 4.59 | 2438 | | | |
| | 75 | 7.65 | 967 | | | |
| | 105 | 10.71 | 195 | | | |
| 16 h | 15 | 1.53 | 8402 | | | |
| | 45 | 4.59 | 2631 | | | |
| | 75 | 7.65 | 990 | | | |
| | 105 | 10.71 | 210 | | | |
| 24 h | 15 | 1.53 | 8420 | | | |
| | 45 | 4.59 | 2638 | | | |
| | 75 | 7.65 | 970 | | | |
| | 105 | 10.71 | 195 | | | |

## 4. Results and Discussion

*4.1. Changes in the Permeability Characteristics of the Geotextile–Polynorbornene Liner over Time after Contacting the Pollutant*

Table 2 reports the test results, and Figure 8 shows the corresponding graphs. The test results for each test condition in Table 2 are the mean values of three experiments. In each test condition, the three experimental results had different values. However, the error was ignored because the difference between the values was outside the range of significant figures. When there was no contact with the oil pollutant (pollutant contact time = 0 h), the permeability coefficient ranged from $10^{-3}$ to $10^{-4}$ cm/s depending on the size of the pressure head, which is similar to the flow velocity of groundwater in the weathered granite soils in Korea. Hence, the flow of groundwater was not affected by contact with the geotextile–polynorbornene liner.

**Table 2.** Test results.

| Contact Time of Oil Pollutant | Pressure Head ($\Delta P$, kPa) | Hydraulic Gradient | Flux ($F$, cm$^3$/cm$^2 \cdot$s) | Permeability Coefficient ($k$, cm/s) |
|---|---|---|---|---|
| 0 h | 15 | 77 | $9.68 \times 10^{-2}$ | $1.26 \times 10^{-3}$ |
| | 45 | 230 | $2.11 \times 10^{-1}$ | $9.17 \times 10^{-4}$ |
| | 75 | 383 | $2.88 \times 10^{-1}$ | $7.53 \times 10^{-4}$ |
| | 105 | 536 | $3.46 \times 10^{-1}$ | $6.46 \times 10^{-4}$ |
| 0.5 h | 15 | 77 | $6.22 \times 10^{-4}$ | $8.12 \times 10^{-6}$ |
| | 45 | 230 | $7.65 \times 10^{-4}$ | $3.33 \times 10^{-6}$ |
| | 75 | 383 | $8.29 \times 10^{-4}$ | $2.17 \times 10^{-6}$ |
| | 105 | 536 | $1.11 \times 10^{-3}$ | $2.06 \times 10^{-6}$ |
| 1 h | 15 | 77 | $1.84 \times 10^{-5}$ | $1.24 \times 10^{-6}$ |
| | 45 | 230 | $1.06 \times 10^{-4}$ | $7.64 \times 10^{-7}$ |
| | 75 | 383 | $2.93 \times 10^{-4}$ | $4.61 \times 10^{-7}$ |
| | 105 | 536 | $6.63 \times 10^{-4}$ | $2.41 \times 10^{-7}$ |
| 2 h | 15 | 77 | $1.28 \times 10^{-5}$ | $6.40 \times 10^{-7}$ |
| | 45 | 301 | $9.04 \times 10^{-5}$ | $4.56 \times 10^{-7}$ |
| | 75 | 383 | $1.74 \times 10^{-4}$ | $3.01 \times 10^{-7}$ |
| | 105 | 536 | $3.43 \times 10^{-4}$ | $1.67 \times 10^{-7}$ |
| 4 h | 15 | 77 | $1.21 \times 10^{-6}$ | $9.52 \times 10^{-8}$ |
| | 45 | 230 | $4.08 \times 10^{-6}$ | $2.69 \times 10^{-8}$ |
| | 75 | 383 | $1.03 \times 10^{-5}$ | $1.78 \times 10^{-8}$ |
| | 105 | 536 | $5.10 \times 10^{-5}$ | $1.58 \times 10^{-8}$ |
| 16 h | 15 | 77 | $1.18 \times 10^{-6}$ | $8.84 \times 10^{-8}$ |
| | 45 | 230 | $3.78 \times 10^{-6}$ | $2.63 \times 10^{-8}$ |
| | 75 | 383 | $1.00 \times 10^{-5}$ | $1.65 \times 10^{-8}$ |
| | 105 | 536 | $4.74 \times 10^{-5}$ | $1.55 \times 10^{-8}$ |
| 24 h | 15 | 77 | $1.18 \times 10^{-6}$ | $9.52 \times 10^{-8}$ |
| | 45 | 230 | $3.77 \times 10^{-6}$ | $2.68 \times 10^{-8}$ |
| | 75 | 383 | $1.03 \times 10^{-5}$ | $1.64 \times 10^{-8}$ |
| | 105 | 536 | $5.10 \times 10^{-5}$ | $1.54 \times 10^{-8}$ |

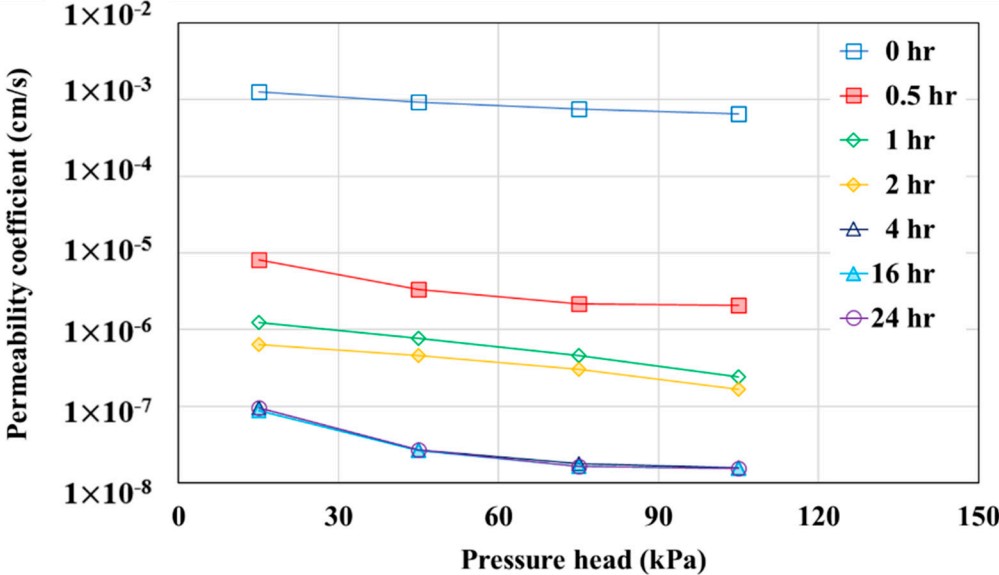

**Figure 8.** Permeability coefficient according to the pressure head.

As shown in Figure 8, changes in the permeability coefficient according to the pressure head were measured at different pollutant contact times. First, regardless of the pollutant contact time, the permeability coefficient decreased as the pressure head increased (the hydraulic gradient increased). When the pollutant contact time was 4 h or longer, the permeability coefficient was $10^{-7}$ cm/s or less, which is defined as almost an impermeable layer. The values showed a tendency to slowly converge when the pressure head was 75 kPa or higher. In other words, the geotextile–polynorbornene liner was likely to further react with the pollutant when the contact time was shorter than 4 h and the pressure head was lower than 75 kPa. However, when the contact time was longer and the pressure head was higher, the reaction between the geotextile–polynorbornene liner and the pollutant was almost complete.

Figure 9 shows the permeability coefficient over time under different pressure head conditions. The permeability coefficient was high when there was no contact with the pollutant, but it sharply decreased at a pollutant contact time of 0.5 h. In addition, regardless of the pressure head, the permeability coefficient did not substantially change once a pollutant contact time of 4 h was reached. Thus, the reaction between the geotextile–polynorbornene liner and the pollutant was completed after 4 h of contact. This result indicates that it is possible to form an impervious layer that can block pollutants.

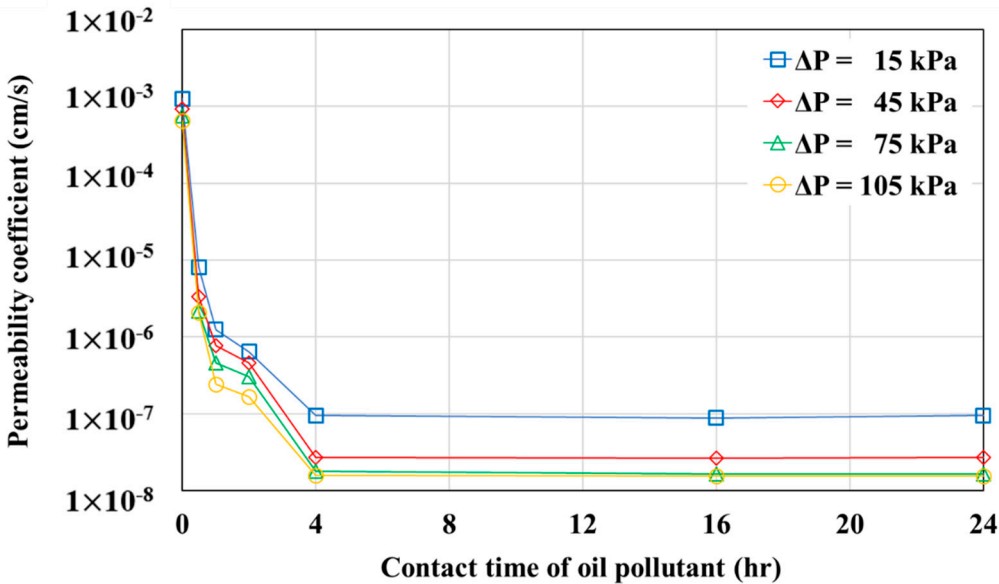

**Figure 9.** Permeability coefficient according to contact time.

Changes in the permeability characteristics of the geotextile–polynorbornene liner mentioned in this section were measured by applying diesel as an oil pollutant. As oil spills on the ground occur at various concentrations, changes in the permeability characteristics of the reactive material must be analyzed at different oil concentrations to evaluate its applicability. Therefore, a three-dimensional (3D) numerical analysis was conducted to simulate different oil concentrations, and the results are presented in Section 4.2.

### 4.2. Numerical Analysis

#### 4.2.1. Finite Difference Analysis (FDA)

In this study, FDA was conducted using the well-known environmental simulation software MT3D (Visual MODFLOW; USGS, Denver, CO, USA) to analyze changes in the permeability characteristics of the geotextile–polynorbornene liner resulting from different concentrations of the oil pollutant. MT3D facilitates the 3D FDA of a hydraulic model for solute movement in a complicated hydrogeological structure. This software has been widely used for pollutant diffusion analysis because it can account for the steady-state flow,

transient flow, anisotropic dispersion, first-order decay, chemical reactions between solutes, and linear and nonlinear adsorption.

Figure 10 shows the analysis model implemented in 3D and its plane view. The analysis model was composed of an oil tank that can generate the pressure head of the oil pollutant, soils with a permeability coefficient of $10^{-4}$ cm/s, and the geotextile–polynorbornene liner in the soils. For the mesh in the analysis, a square of 0.1 m was used for the oil tank and soils. The thickness of the geotextile–polynorbornene liner was 0.06 m. In addition, pollutant monitoring wells at four locations were simulated to examine the concentration of pollutants that passed through the geotextile–polynorbornene liner. The dimensions of the oil tank and soils were set to $0.24 \times 0.5 \times 1.2$ m and $1.0 \times 0.5 \times 0.5$ m (L × W × H). It is worth noting that a soil box can be used in further research.

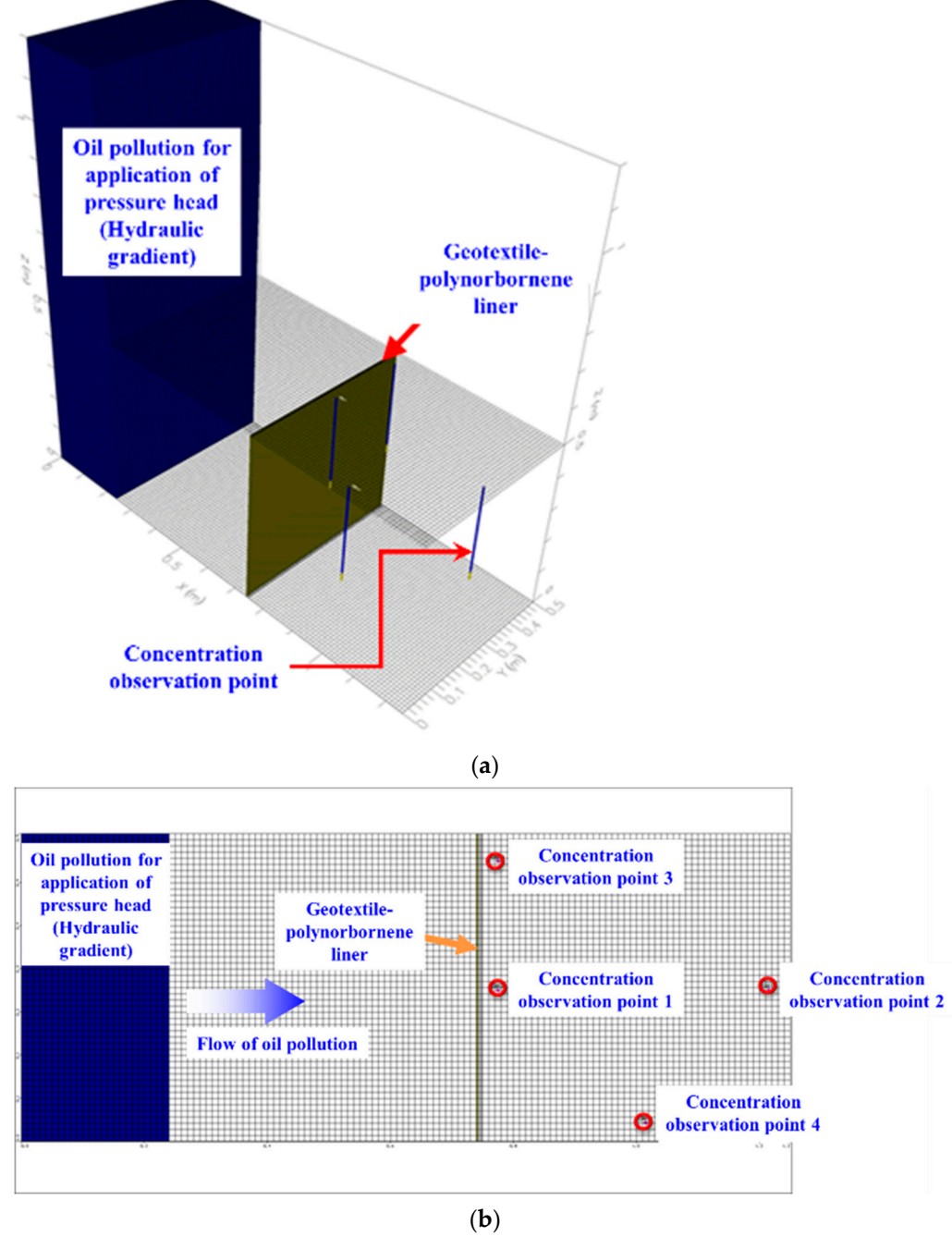

**Figure 10.** The 3D FDA model: (**a**) 3D FDA model; (**b**) plane view of the analysis model.

The hydraulic conductivity represents the degree of smoothness of the liquid material flow in the soil. A higher hydraulic conductivity indicates a smoother flow of the liquid material. Therefore, the hydraulic conductivity of the oil tank was set to 1000 cm/s so that the oil could be smoothly introduced to the simulated soils. In addition, $10^{-4}$ cm/s was applied as the permeability coefficient of the soils. Table 3 lists the conditions for the analysis model.

**Table 3.** FDA model conditions.

| Classification | Oil Tank | Soils |
|---|---|---|
| Porosity | 0.9 | 0.25 |
| Horizontal permeability coefficient (cm/s) | 1 | $10^{-4}$ |
| Vertical permeability coefficient (cm/s) | 1 | $10^{-4}$ |
| Specific storativity (m$^{-1}$) | $10^{-5}$ | $10^{-5}$ |
| Specific yield | 0.9 | 0.15 |
| Contact time of oil pollutant (h) | 96 | |

In general, the processes that govern the transport of pollutants include groundwater flow, pollutant adsorption, advection, diffusion, dispersion, and biodegradation. The purpose of this study, however, was to examine the impermeability performance of the geotextile–polynorbornene liner when oil pollutants with different concentrations are released in soils. Therefore, only the influences of advection, diffusion, and dispersion were considered for the prediction of pollutant movement by FDA. TPH, which can simulate diesel, was applied as the pollutant type. In addition, because the coefficient results show that permeability changed to impermeability as the pollutant contact time increased from 0.5 to 4 h, the permeability coefficients obtained when the geotextile–polynorbornene liner was in contact with the pollutant for 0.5 and 4 h were applied in the FDA. Preliminary analysis confirmed that a pressure head of 15 kPa was too small to affect the FDA results. Therefore, 45, 75, and 105 kPa were applied as pressure head conditions. Table 4 lists the FDA cases.

**Table 4.** FDA cases.

| Analysis Cases | Pollutant (TPH) Concentration (ppm) | Contact Time of Pollutant (h) | Pressure Head ($\Delta P$, kPa) | Permeability Coefficient of Geotextile–Polynorbornene Liner (cm/s) |
|---|---|---|---|---|
| Case HC-1 | | | 45 | $3.33 \times 10^{-6}$ |
| Case HC-2 | | 0.5 | 75 | $2.17 \times 10^{-6}$ |
| Case HC-3 | 6000 | | 105 | $2.06 \times 10^{-6}$ |
| Case HC-4 | | | 45 | $2.69 \times 10^{-8}$ |
| Case HC-5 | | 4 | 75 | $1.78 \times 10^{-8}$ |
| Case HC-6 | | | 105 | $1.58 \times 10^{-8}$ |
| Case MC-1 | | | 45 | $3.33 \times 10^{-6}$ |
| Case MC-2 | | 0.5 | 75 | $2.17 \times 10^{-6}$ |
| Case MC-3 | 2000 | | 105 | $2.06 \times 10^{-6}$ |
| Case MC-4 | | | 45 | $2.69 \times 10^{-8}$ |
| Case MC-5 | | 4 | 75 | $1.78 \times 10^{-8}$ |
| Case MC-6 | | | 105 | $1.58 \times 10^{-8}$ |
| Case LC-1 | | | 45 | $3.33 \times 10^{-6}$ |
| Case LC-2 | | 0.5 | 75 | $2.17 \times 10^{-6}$ |
| Case LC-3 | 500 | | 105 | $2.06 \times 10^{-6}$ |
| Case LC-4 | | | 45 | $2.69 \times 10^{-8}$ |
| Case LC-5 | | 4 | 75 | $1.78 \times 10^{-8}$ |
| Case LC-6 | | | 105 | $1.58 \times 10^{-8}$ |

### 4.2.2. Changes in the Permeability Characteristics of the Geotextile–Polynorbornene Liner According to the Concentration of the Oil Pollutant

Figures 11–13 show the concentration of the pollutant at each observation point over time after the pollutant passed through the geotextile–polynorbornene liner at different TPH concentrations.

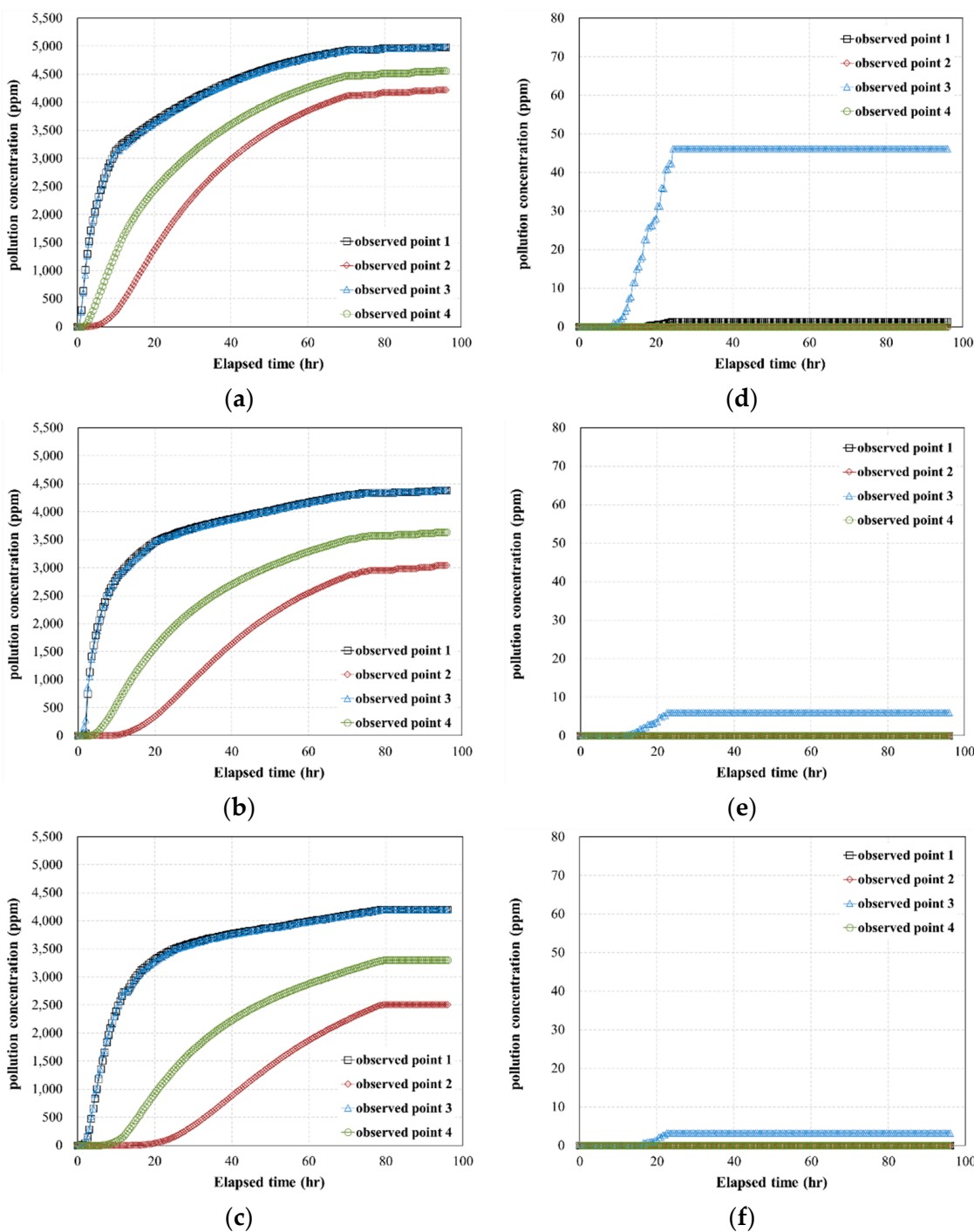

**Figure 11.** FDA results for the high concentration (6000 ppm) condition: (**a**) HC-1; (**b**) HC-2; (**c**) HC-3; (**d**) HC-4; (**e**) HC-5; (**f**) HC-6.

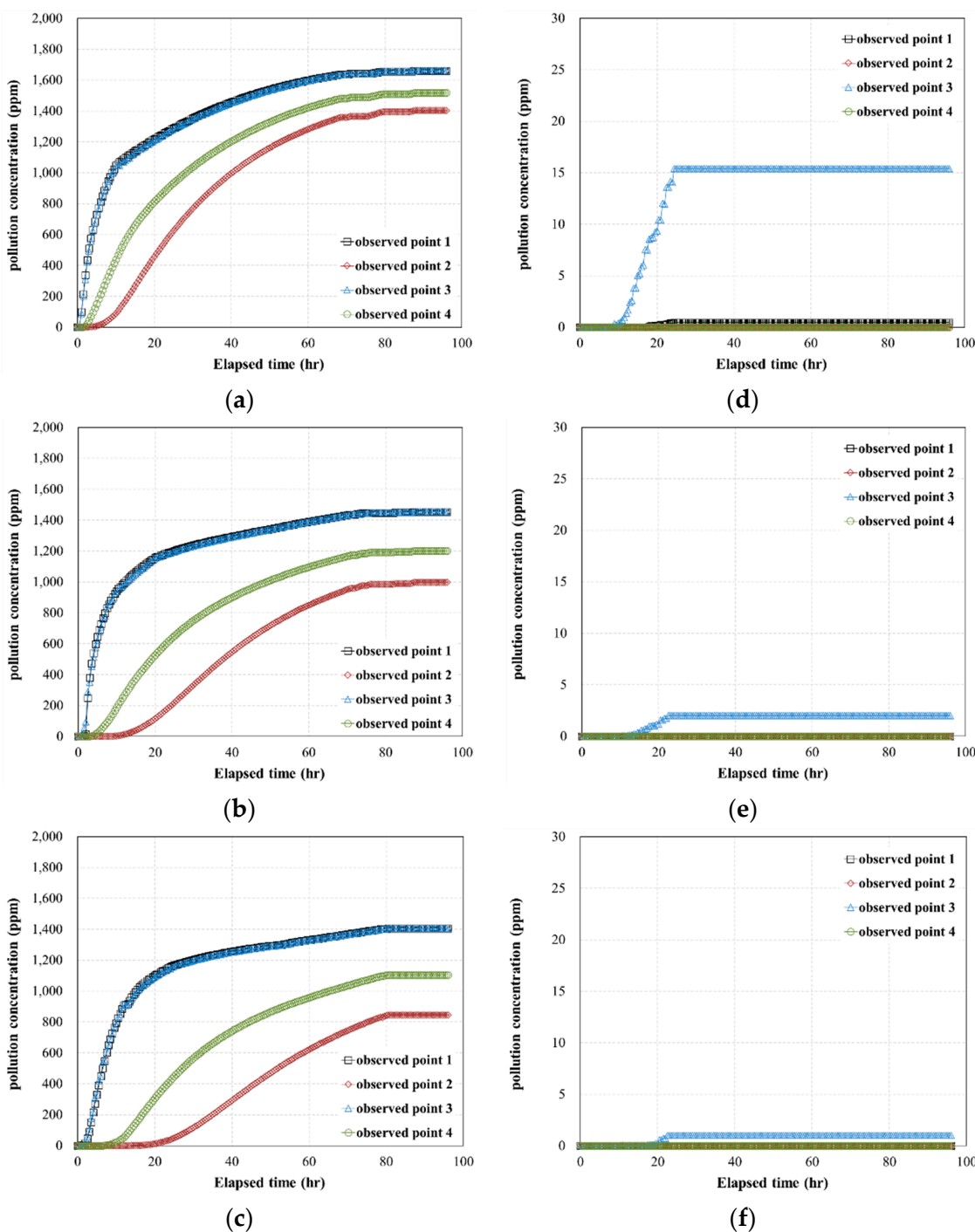

**Figure 12.** FDA results for the moderate concentration (2000 ppm) condition: (**a**) MC-1; (**b**) MC-2; (**c**) MC-3; (**d**) MC-4; (**e**) MC-5; (**f**) MC-6.

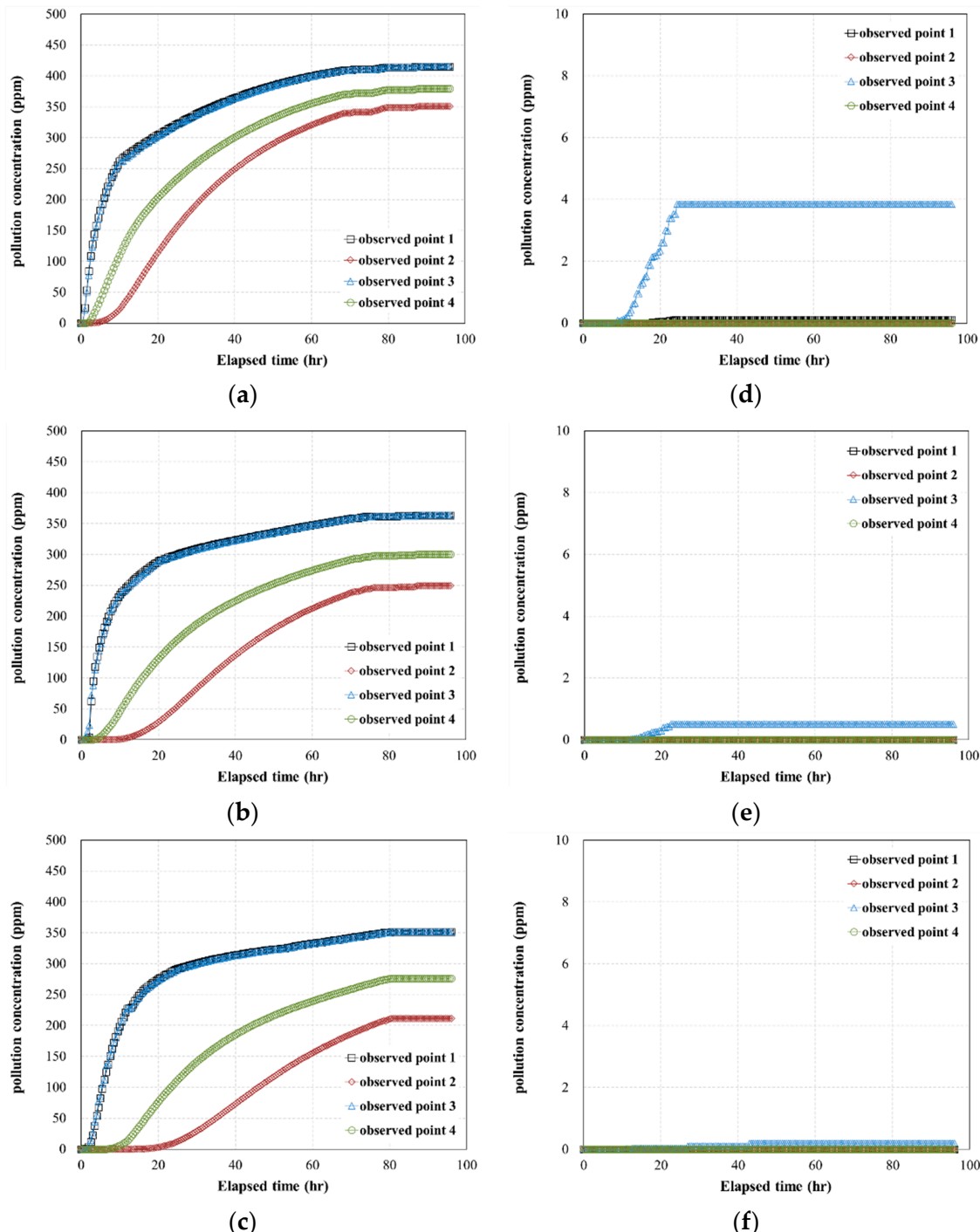

**Figure 13.** FDA results for the low concentration (500 ppm) condition: (**a**) LC-1; (**b**) LC-2; (**c**) LC-3; (**d**) LC-4; (**e**) LC-5; (**f**) LC-6.

Figure 11a–c (cases HC-1–HC-3) shows the concentration change over time for 0.5 h of contact between the pollutant and the geotextile–polynorbornene liner. As the permeability coefficient of the geotextile–polynorbornene liner decreased, the concentration of the pollutant tended to decrease at each observation point, but the concentration continued to increase over time. In addition, for each permeability coefficient, a higher concentration of the pollutant was released at the observation points adjacent to the geotextile–polynorbornene liner, and the pollutant concentration significantly increased within a short period of time. As the distance from the geotextile–polynorbornene liner increased, however, the pollutant

concentration decreased, and the rate of pollutant increase was lower than that in the adjacent observation points.

Figure 11d–f (cases HC-4–HC-6) shows the concentration change over time for a contact time of 4 h between the pollutant and the geotextile–polynorbornene liner. As the permeability coefficient of the geotextile–polynorbornene liner decreased, the concentration of the pollutant tended to decrease at each observation point. In addition, the pollutant concentrations at the observation points adjacent to the geotextile–polynorbornene liner were higher than those at the observation points far away from it. However, the change in concentration with the pollutant contact time significantly decreased for the same permeability coefficient and observation point. Furthermore, at most observation points, the pollutant concentration did not substantially change over time. This is the result of the impermeability effect of the geotextile–polynorbornene liner as well as the influence of the distances of the observation points.

The same tendencies of the analysis results mentioned above were observed at moderate (Figure 12) and low concentrations (Figure 13). Under each analysis condition, the maximum pollutant concentration was the highest at the observation point that was closest to the geotextile–polynorbornene liner, as shown in Table 5.

**Table 5.** Maximum pollutant concentration at the observation points adjacent to the geotextile–polynorbornene liner.

| Analysis Cases | Pollutant (TPH) Concentration (ppm) | Contact Time of Pollutant (h) | Pressure Head ($\Delta P$, kPa) | Maximum Concentration of Observed Point 1 (ppm) |
|---|---|---|---|---|
| Case HC-1 | | | 45 | 4985.8 |
| Case HC-2 | | 0.5 | 75 | 4379.6 |
| Case HC-3 | | | 105 | 4200.9 |
| Case HC-4 | 6000 | | 45 | 46.1 |
| Case HC-5 | | 4 | 75 | 6.0 |
| Case HC-6 | | | 105 | 3.2 |
| Case MC-1 | | | 45 | 1660.5 |
| Case MC-2 | | 0.5 | 75 | 1453.2 |
| Case MC-3 | | | 105 | 1405.3 |
| Case MC-4 | 2000 | | 45 | 15.4 |
| Case MC-5 | | 4 | 75 | 2.0 |
| Case MC-6 | | | 105 | 1.0 |
| Case LC-1 | | | 45 | 415.13 |
| Case LC-2 | | 0.5 | 75 | 363.31 |
| Case LC-3 | | | 105 | 351.32 |
| Case LC-4 | 500 | | 45 | 3.8 |
| Case LC-5 | | 4 | 75 | 0.5 |
| Case LC-6 | | | 105 | 0.2 |

At an initial pollutant concentration of 6000 ppm and a contact time of 0.5 h, the ratio of the concentration of the pollutant that permeated through the geotextile–polynorbornene liner to the initial pollutant concentration ranged from 70.02 to 83.1%. For a pollutant contact time of 4 h, however, the concentration ranged from 0.08 to 0.92% compared to that at 0.5 h, and the ratio ranged from 0.05 to 0.77%. When the initial pollutant concentrations were 2000 and 500 ppm, the concentrations of the pollutant that permeated through the geotextile–polynorbornene liner over time were similar to the results for 6000 ppm. In other words, the numerical analysis results show that the geotextile–polynorbornene liner has a pollutant blocking effect over time.

## 5. Conclusions

In this study, changes in the permeability characteristics of an oil-absorbing medium were tested, and experiments and numerical analysis were used to evaluate a geotextile–

polynorbornene liner for its ability to prevent the diffusion of pollutants. The results are as follows:

1.  When changes in the permeability coefficient were examined at different pressure heads and different pollutant contact times, the permeability coefficient decreased as the pressure head increased (the hydraulic gradient increased) regardless of the pollutant contact time. In addition, when the pollutant contact time was 4 h or longer, the permeability coefficient of the geotextile–polynorbornene liner was $10^{-7}$ cm/s or less, which is defined as almost an impermeable layer.

2.  Changes in the permeability coefficient were examined over time under different pressure head conditions. There was almost no change in the permeability coefficient starting from the pollutant contact time of 4 h. Thus, when the pollutant contact time reaches 4 h or more, the geotextile–polynorbornene liner has an impermeable layer that can block pollutants.

3.  The results of the 3D pollutant diffusion analysis showed that, for a pollutant contact time of 4 h, the maximum concentration of the pollutant that permeated through the geotextile–polynorbornene liner was less than approximately 0.8% compared to the initial pollutant concentration. Therefore, the numerical analysis results confirm that the geotextile–polynorbornene liner has a pollutant blocking effect over time.

4.  The test and numerical analysis results confirm the impermeability performance of the geotextile–polynorbornene liner against oil pollutants. Therefore, it has potential as an application for the prevention of pollutant diffusion.

**Funding:** This research received no external funding.

**Institutional Review Board Statement:** Not applicable.

**Informed Consent Statement:** Not applicable.

**Data Availability Statement:** The data presented in this study are available on request to the corresponding author. The data are not publicly available as they form part of an ongoing study.

**Conflicts of Interest:** The author declares no conflict of interest.

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
