# Peer review of "Evaluation of Changes in the Permeability Characteristics of a Geotextile–Polynorbornene Liner for the Prevention of Pollutant Diffusion in Oil-Contaminated Soils"

_sustainability, doi:10.3390/su13094797_

Round 1

Reviewer 1 Report

The authors should considers submitting the paper to a geotechnical area based journal for better impact of the paper.

Reviewer 2 Report

The permeability alteration characteristics of the oil-absorbing medium 403 were tested and evaluated for its ability and applicability to prevent the diffusion of pol-404 lutants using test methods and numerical analysis. The manuscript is well organized and written.

several parts to improve:

1) Figure 6 doesnt contain scale bar

2) For cross section image, SEM and element distribution should be added.

3) For the test results, the mean and error should also include., e.g. table 2. 

Reviewer 3 Report

Comments to Park

Summary

The manuscript treats a particular form of barrier against oil pollution assessing its permeability depending on contamination contact time through simulation. The barrier, which under normal conditions allows the passage of water, contains a material that swells when in contact with oil preventing its penetration.

General comments

The topic of the study is relevant for the Sustainability journal and the results have general interest in the realm of environmental protection. Furthermore, the manuscript contains the customary elements of a scientific paper as well as a separate section with the title Overview of Geotextile-Polynorbornene Linear between the Introduction and Material and Method sections describing the barrier material. As for the substance of the text, the authors give a reasonable description of the simulation work and the subsequent conclusions, but I wish there were more context to the employed barriers and the results. Where does the geotextile-polynorbornene linear come from? Is it an innovation? How does it perform compared to other barriers? How is it linear?

The English of the manuscript is rather good, but the presentation could be more straightforward; a scientific text, unlike a poem, does not need ornaments, but the emphasis should be on the clarity.

Specific comments

Line 52: I think the conjunction while must appear in a sentence with two clauses separated by a comma. (A behaves in one way, while B does otherwise.) It cannot initiate a stand-alone sentence. Hence grammatically, there should be comma between the words site and while instead of period. I however admit that the sentence then would be unnecessarily long. Therefore, it is better to replace while with e.g. on the other hand.   

Line 261: For the units of the both sides of Equation (1) to agree, Q should be flow rate (m3/s).

Equation (2): What does the notation V(F) mean? Should it be just F on the left-hand side?

Line 263: Usually we give the pressure head as a length (like the mm Hg in a barometer). We can of course convert it to the corresponding pressure, but the convention is that pressure head is the height of the liquid column that exerts a pressure.

Equation (4): Again, problems with the units. What is the unit of Δh on the left-hand side? If it is a head, it should be a length, but how does it then match the unit on the right-hand side? I suspect that your ΔP is just a pressure and then a modified version of Equation (4) converts it to the pressure head Δh.

Δh = ΔP/(g•ρ)

In the modified equation g is the gravitational acceleration.

Table (1): Why are most of the entries empty in the three last columns?

Figure 11: Here and in subsequent figures the x-axis label should be elapsed time.
